# Intention to obtain a COVID-19 vaccine among Brazilian immigrant women in the U.S.

**Jennifer D. Allen**[1]*, **Leticia Priebe Rocha**[1], **Raviv Rose**[1], **Annmarie Hoch**[1], **Thalia Porteny**[2], **Adriana Fernandes**[3], **Heloisa Galvão**[4]

**1** Department of Community Health, Tufts University, Medford, MA, United States of America, **2** Department of Occupational Health, Tufts University, Medford, MA, United States of America, **3** Office of Immigrant Affairs, City of Somerville, Somerville, MA, United States of America, **4** Brazilian Women's Center, Brighton, MA, United States of America

\* Jennifer.allen@tufts.edu

## Abstract

### Background

COVID-19 has disproportionately impacted low-income immigrant communities. There is concern that the current uptake of COVID-19 vaccines is suboptimal and that this may be contributing to COVID-19 inequities. However, little is known about the acceptability of COVID-19 vaccines among immigrants in the U.S. Our goal was to gauge COVID-19 vaccine intentions among Brazilian immigrant women living in the U.S.

### Methods

We conducted an online survey between July and August 2020 offered in Portuguese and English languages among a convenience sample of Brazilian immigrant women ages 18 years and older. Women were recruited through online advertisements by community-based organizations and social media groups to complete a survey that assessed intention to get a COVID-19 vaccine, attitudes toward vaccines, and perceptions about the pandemic.

### Results

Of the total sample (N = 353), most (70.8%) indicated they intended to get a COVID-19 vaccine. In bivariate analyses, vaccine intentions were significantly associated with perceptions about the severity of the pandemic, trusted sources of health information, and the number of years lived in the U.S. Multinomial logistic regression models revealed that those who did not intend to be vaccinated had lived a longer time in the U.S. (OR: -0.12 95% CI: -0.19, -0.05), perceived the pandemic to be a minor issue (OR: 1.52, 95% CI: 0.62, 2.42), and trusted information from social networks (OR: -1.94, 95% CI: -3.25, -0.63) or private news sources (OR: -1.71, 95% CI: -2.78, -0.63).

### Conclusions

While most women reported they would get a COVID-19 vaccine, efforts to reach those who may be hesitant should target those who have lived in the U.S. for longer periods of time and

**Data Availability Statement:** The dataset for this study can be found at the following link from the Open ICPSR database. https://doi.org/10.3886/E164382V1.

**Funding:** Funding for this study was provided to JA in part by a grant from the Tisch College Community Research Center (https://tischcollege.tufts.edu/research/tcrc). Additional funding was provided by the National Center for Advancing Translational Sciences, National Institutes of Health, Award Number UL1TR002544 (https://ncats.nih.gov/). The content is solely the responsibility of the authors and does not necessarily represent the official views of the NIH. The funders had no role in study design, data collection and analysis, decision to publish, or preparation of the manuscript.

**Competing interests:** The authors have declared that no competing interests exist.

do not perceive the pandemic to be a major crisis. Healthcare providers may be particularly suited to deliver this information given high levels of trust.

## Introduction

The COVID-19 pandemic has devastated communities across the world, completely shifting our way of life at an unforeseen human cost. It pandemic has worsened long-standing health inequities in the U.S., with a disproportionate number of cases and deaths among Black people and non-Black Latinos [1–3]. Although these groups represent only approximately 30% of the general U.S. population, 55% of COVID-19 cases have occurred among people who are Latino and Black [4]. In a study of community-level factors associated with disparities in COVID-19 incidence, communities with a higher proportion of foreign-born non-citizens had more COVID-19 cases than those with lower proportions of immigrants [5]. Various commentaries further highlight the disproportionate burden of COVID-19 among immigrant communities and call for the prioritization of vaccine access within these vulnerable communities [6–11], yet little is currently known about COVID-19 vaccine acceptability among these populations.

In this study, we focus on Brazilian immigrants, both because they represent a growing number of immigrants to the U.S. and because there is little data available on their health status. In 2017, it was estimated that there were more than 450,000 Brazilian immigrants living in the U.S., although this may be inaccurate because a significant proportion of the Brazilian immigrant population do not have legal documents and may be underrepresented in the Census [12]. Indeed, the Brazilian Consulate-General in Boston, MA estimates that there are 350,000 Brazilians living within their jurisdiction alone [12, 13]. Another aspect of the relative "invisibility" of Brazilians is that they are categorized in the U.S. as Hispanic (e.g., Spanish-speaking), but few Brazilians identify with this label since they speak Portuguese. Some do not even identify with the term "Hispanic," in part because Brazilian culture has strong African, Indigenous, and Portuguese influences, rather than Spanish influences [13, 14]. Thus, even as information about the impact of COVID-19 among immigrant populations becomes available, it may be difficult to assess the impact among Brazilians as a distinct ethnic group. This is of concern as Brazilians living in the U.S. are over-represented in low-wage, public-facing jobs that frequently lack paid sick leave or opportunities to work from home, putting them at greater risk for exposure to COVID-19 [1, 6–8, 10, 11, 15]. Like many immigrant communities, they also face myriad barriers to accessing healthcare, including lack of insurance and low English language proficiency [10].

The goal of this study was to assess COVID-19 vaccine intentions among Brazilian immigrants and to gather information that may inform interventions to control the spread of COVID-19. We chose to focus on Brazilian women because they are often the gatekeepers to healthcare for their families, arranging medical appointments for their children and partners [16, 17]. Additionally, at the time this study was conducted, there were studies and polls indicating that women may be less likely to intend to be vaccinated for COVID-19 than men [18]. This community-engaged study was conducted by investigators at Tufts University, the Brazilian Women's Group (brazilianwomensgroup.org), and the Somerville Office of Immigrant Affairs.

## Materials and methods

The Health Belief Model provides the conceptual underpinnings for this study [19]. Specifically, the model predicts that behavior is related to an individual's perceived threat of an illness

(perceived susceptibility), beliefs about its consequences (perceived severity), positive benefits and barriers to action, as well as exposure to factors that prompt action (cues to action), and confidence in the ability to succeed (self-efficacy). According to the theory, an individual that perceives a threat to health, is consecutively cued to action, and perceives that the benefits to action outweigh the barriers, are more likely to undertake the recommended preventive health action. In the current study, we conceptualize perceived susceptibility as having had a positive COVID test, perceived severity as subjective perceptions regarding the seriousness of the pandemic (particularly since the information from the Brazilian president suggested otherwise), perceived barriers (e.g., health insurance, limited English language proficiency), and experiencing cues to action through trusted information sources. We also considered factors that have been found in empirical studies to impact beliefs about COVID vaccination, such as age, marital status, household income [20–22], as well as time spent living in the U.S. [23–25].

## Sample and setting

Data for this analysis are from a larger study on Brazilian immigrant women's health priorities and experiences. A convenience sample was recruited from Brazilian social media pages (e.g., Facebook, WhatsApp) and through outreach by collaborating community partners (e.g., health and social service providers in Boston, MA) who posted information and a link to the study on their social media pages. Those eligible to participate were women aged 18 years or older, born in Brazil, and reported current residence in the U.S. The online survey was offered in Portuguese and English, and respondents could choose based on their language preference. The survey was translated by an American Translators Association-certified translator and was reviewed by native Portuguese speakers and an expert in Portuguese linguistics to ensure that the translation was linguistically and culturally accurate. Survey respondents were provided with a financial incentive ($20 gift card) for survey completion, which took an average of 18.5 minutes. Data was collected between July and August 2020. Potential respondents were required to read and consent to study participation before by checking a box on the survey cover page. Consent information included information about the study purpose, topics addressed in survey questions, and the voluntary nature of participation. All procedures were approved by the Institutional Review Board at Tufts University, Medford, MA, USA (protocol number 00001838).

## Measures

We assessed our primary outcome (COVID-19 vaccine intentions) by asking: "If a vaccine became available to prevent the Coronavirus, would you want to get it?" (yes/no/don't know). For those who responded that they would not get the vaccine we asked, "Why not?" and respondents were able to enter free-text responses. We also assessed prior testing for and diagnosis of COVID-19 ("Have you been tested for the Coronavirus? If so, what was the result?)" with response options: "I have been tested and I tested positive (I had coronavirus)," "I have been tested and I tested negative (I did not have coronavirus)," "I have been tested and I do not know the result," or "I have not been tested." We assessed perceptions regarding the seriousness of the pandemic with a question asking respondents whether they thought that the pandemic was "a significant crisis," "a serious problem but not a crisis," "a minor problem," or "not a problem at all" [18].

　　We assessed trusted sources of health information by asking: "What source do you trust the most to give you accurate up-to-date information about health?" with response options: healthcare provider (i.e. doctor/physician, nurse practitioner/nurse); local network news; social network members (i.e., family member or friends); social media; public health agencies

(e.g., Centers for Disease Control and Prevention); internet sites (e.g., WebMD, Google); governmental agencies or officials; pharmaceutical companies; religious or faith leaders.

Socio-demographic characteristics were assessed using items from the Brazilian Census [26], including race/ethnicity (White/Black/Indigenous/Asian/Pardo ["mixed"]), and educational attainment (incomplete primary education/completed primary/complete secondary (high school diploma) school and incomplete tertiary education (college degree)/complete tertiary education). We also assessed age, household income, and insurance status using items from the Behavioral Risk Factor Surveillance System [27]. We inquired about nativity with the question: "What is your country of birth" (Brazil/United States/ other). We asked about the number of years lived in the U.S. since prior evidence suggests that longer time spent living in the U.S. is associated with the adoption of health behaviors that are normative in this country [23–25]. Moreover, given that vaccine availability differs by geographic region, we asked the state of residence. Additionally, since acculturation appears to change beliefs, attitudes, and values regarding health behaviors and healthcare among immigrants [28], we asked about language spoken at home.

## Analysis

We eliminated respondents who were born in a country other than Brazil (n = 11) and who had missing data on vaccine intentions (n = 1), leaving a final analytic sample of N = 353. For respondents that did not complete questions on household income and trusted information sources, we categorized these responses as "missing," and retained these respondents in the sample since we did not want to lose information for the overall analysis.

Before analysis, we combined categories for some variables due to small cell sizes (n<10). Specifically, we re-categorized answers for marital status (single, never married/married or living as married/formerly married or separated), responses to the item regarding perceived significance of the pandemic were combined into two categories: "minor crisis" (a serious problem but not a crisis, a minor problem, not a problem at all) versus "a significant crisis" relabeled as "major crisis," and most trusted sources of health information were combined: doctor/physician and nurse practitioner/nurse combined to "healthcare provider"; public health agencies and governmental agencies were combined in "public agencies"; social media and family members or friends combined to a category entitled "social networks"; internet and network news were combined to "private news sources".

Descriptive statistics, including percentages, means, standard deviations, and ranges, were produced for all variables. We evaluated bivariate associations between vaccination intentions (outcome), key independent variables (e.g., COVID-19 experiences and perceptions, trusted sources of information) with Chi-squares and ANOVA.

For multivariable analyses, we combined those who did not intend to be vaccinated with those who were unsure about vaccination, due to sample size and prior evidence that those who refuse and those who are unsure about COVID-19 vaccination are similar in terms of correlates [29, 30]. Variables selected for inclusion in multivariable analyses included our key independent variables (prior COVID infection, perceived significance of the pandemic, trusted sources of health information), covariates deemed important predictors of COVID-19 vaccine intentions in prior research (e.g., age, income) [20–22], as well as other sociodemographic characteristics that demonstrated statistically significant associations with vaccine intentions in bivariate analyses ($p < 0.05$). Statistical significance was considered at the $p < 0.05$ level for the final model. Quantitative data analysis was generated using Stata software [31].

Analysis of free-text responses (reasons for not wanting to be vaccinated) was done using thematic analysis [32, 33]. Based on published literature on vaccine hesitancy [34], we

categorized responses in terms of vaccine concerns (e.g., the vaccine has not been fully tested, could have serious side effects, was not effective) and trust in authorities charged with developing and/or administering the vaccine.

# Results

## Characteristics of study participants

A total of 353 Brazilian women were included in the analysis, with most (92.6%) completing the survey in Portuguese. The mean age was 39.4 years (SD = 11.8). Approximately two-thirds (63%) reported their race as White, and a quarter (25.8%) reported themselves as Pardo (mixed-race). Notably, a majority (78.2%) identified as "Latina," yet only 5.4% identified as "Hispanic," which supports previous reports that Brazilians do not identify as Hispanic [13, 14, 35]. The average time living in the U.S. was 11.81 years (SD = 9.09, range = 0.25–40). Most lived in Massachusetts (70.4%), with smaller percentages residing in New Jersey (6.3%), Florida (5.7%), and California (5.1%). Nearly half of participants (47.0%) spoke only Portuguese at home and less than half (44.2%) spoke both English and Portuguese at home. Over half (512. %) reported an annual household income of less than $50,000. Nearly half (48.7%) had completed tertiary education (equivalent to a U.S. college degree). The majority (68.8%) reported being married or living as married. More than a third (36.4%) had public insurance and 19.3% were uninsured. The majority (72.0%) reported that they had not been tested for COVID-19; only 4.5% reported they had both been tested for and diagnosed with COVID-19. See Table 1.

## COVID-19 vaccine intentions

Most (70.8%) said they would get a COVID-19 vaccine and 18.7% were unsure. Among those who reported that they would not be vaccinated, the major factors (from free text responses) were: the vaccine had not been fully tested (30.9%), could have serious side effects (17.6%), or was not effective (8.8%). Other participants reported that they would not get vaccinated due to mistrust of the government (8.8%) and other systems supporting the production of the vaccine (10.3%), as well as a general mistrust of vaccines (e.g., "I've never taken a vaccine in my life") (8.8%).

## Bivariate results

Participants who considered the pandemic to be a minor problem were less likely to intend to be vaccinated compared to those who considered it a significant crisis (52.6% vs. 76.0%, p < 0.001). Those whose most trusted sources of health information were private news sources or social networks were less likely to report that they would get the vaccine compared to those who listed their most trusted source of information as healthcare providers (63.4% vs. 73.5%). Additionally, those living in the U.S. for longer periods were less likely to intend to be vaccinated than those that had immigrated more recently. See Table 1.

## Multivariate analysis

In Model 1, which included socio-demographic characteristics (age, marital status, income), participants were significantly less likely to intend to get a vaccine if they had spent more time living in the U.S. (OR: -0.10, 95% CI: -0.16, -0.04). In Model 2, which controlled for age, marital status, income, and time in the U.S., we found that both perceived severity of the pandemic and trusted source of health information were associated with vaccine intentions. Specifically, participants who perceived the pandemic to be a major crisis were more likely to say they would be vaccinated compared to those who identified the pandemic as a minor crisis (OR: 1.52, 95% CI: 0.62, 2.42). Additionally, those who reported that their most trusted sources of

**Table 1. Characteristics of study sample, by COVID-19 vaccine intentions (N = 353).**

| | Total (N = 353) | Intend to be vaccinated (N = 250; 70.8%) | Do not intend to be vaccinated (N = 37; 10.5%) | Unsure about vaccination (N = 66; 18.7%) | p-value |
|---|---|---|---|---|---|
| **Socio-demographic characteristics** | N | N (%) | N (%) | N (%) | |
| **Age** (mean = 39.4, SD = 11.8) | | | | | 0.07 |
| 18–34 | 121 | 95 (78.5%) | 6 (7.0%) | 20 (16.5%) | |
| 35–64 | 223 | 150 (67.3%) | 30 (13.5%) | 43 (19.3%) | |
| 65+ | 8 | 5 (62.5%) | 1 (12.5%) | 2 (25.0%) | |
| Missing | 1 | 0 | 0 | 1 (100%) | |
| **Race** (select all that apply) | | | | | N/A |
| White | 222 | 160 (72.1%) | 21 (9.5%) | 41 (18.5%) | |
| Black | 31 | 22 (71.0%) | 5 (16.1%) | 4 (12.9%) | |
| Indigenous | 8 | 6 (75.0%) | 2 (25.0%) | 0 | |
| Asian | 5 | 4 (80.0%) | 1 (20.0%) | 0 | |
| Pardo | 91 | 63 (69.2%) | 9 (9.9%) | 19 (20.9%) | |
| Another race | 21 | 14 (66.7%) | 3 (14.3%) | 4 (19.0%) | |
| Missing | 1 | 1 (100%) | 0 | 0 | |
| **Hispanic** | | | | | 0.54 |
| Yes | 19 | 12 (63.2%) | 4 (21.1%) | 3 (15.8%) | |
| No | 319 | 228 (71.5%) | 32 (10.0%) | 59 (18.5%) | |
| Unsure | 15 | 10 (66.7%) | 1 (6.7%) | 4 (26.7%) | |
| Missing | 0 | 0 | 0 | 0 | |
| **Latina** | | | | | 0.52 |
| Yes | 276 | 196 (71.0%) | 30 (10.9%) | 50 (18.1%) | |
| No | 49 | 34 (69.4%) | 3 (6.1%) | 12 (24.5%) | |
| Unsure | 27 | 20 (74.1%) | 4 (14.8%) | 3 (11.1%) | |
| Missing | 1 | N/A | N/A | N/A | |
| **Nativity** | | | | | N/A |
| Brazil | 353 | 250 (70.8%) | 37 (10.5%) | 66 (18.7%) | |
| Missing | 0 | 0 | 0 | 0 | |
| **Time in U.S** (mean = 11.81, SD = 9.09) | | | | | **0.002** |
| 0–4 years | 106 | 89 (84.0%) | 4 (3.8%) | 13 (12.3%) | |
| >4–9 years | 72 | 52 (72.2%) | 4 (5.6%) | 16 (22.2%) | |
| >9–19 years | 89 | 56 (62.9%) | 15 (16.9%) | 18 (20.2%) | |
| >19–40 years | 86 | 53 (61.6%) | 14 (16.3%) | 19 (22.1%) | |
| Missing | 0 | 0 | 0 | 0 | |
| **Language Spoken at Home** | | | | | 0.42 |
| Portuguese only | 166 | 119 (71.7%) | 13 (7.8%) | 34 (20.5%) | |
| English only | 26 | 15 (57.7%) | 4 (15.4%) | 7 (27.0%) | |
| Portuguese & English | 156 | 112 (71.8%) | 19 (12.2%) | 25 (16.0%) | |
| Other | 5 | 4 (80.0%) | 1 (20.0%) | 0 | |
| Missing | 0 | 0 | 0 | 0 | |
| **Household Income** | | | | | 0.63 |
| <10,000 | 62 | 44 (71.0%) | 6 (9.7%) | 12 (19.4%) | |
| $15,000-$25,000 | 56 | 37 (66.1%) | 7 (12.5%) | 12 (21.4%) | |
| $25,001-$50,000 | 63 | 45 (71.4%) | 4 (6.3%) | 14 (22.2%) | |
| $50,001-$75,000 | 59 | 46 (78.0%) | 4 (6.8%) | 9 (15.3%) | |
| $75,001-$100,000 | 40 | 28 (70.0%) | 6 (15.0%) | 6 (15.0%) | |

(*Continued*)

**Table 1.** (Continued)

| | Total (N = 353) | Intend to be vaccinated (N = 250; 70.8%) | Do not intend to be vaccinated (N = 37; 10.5%) | Unsure about vaccination (N = 66; 18.7%) | p-value |
|---|---|---|---|---|---|
| **Socio-demographic characteristics** | N | N (%) | N (%) | N (%) | |
| $100,000+ | 48 | 32 (66.7%) | 9 (18.8%) | 7 (14.6%) | |
| Unsure/Missing | 25 | 18 (72.0%) | 1 (4.0%) | 6 (24.0%) | |
| **Education** | | | | | |
| < Primary education | 24 | 14 (58.3%) | 4 (16.7%) | 6 (25.0%) | |
| Complete primary education | 37 | 24 (64.9%) | 5 (13.5%) | 8 (21.6%) | |
| Complete secondary education | 116 | 87 (75.0%) | 10 (8.6%) | 19 (16.4%) | |
| Complete tertiary education | 172 | 122 (70.9%) | 18 (10.5%) | 32 (18.6%) | |
| Unsure | 4 | 3 (75.0%) | 0 | 1 (25.0%) | |
| Missing | 0 | 0 | 0 | 0 | |
| **Marital Status** | | | | | 0.23 |
| Single, never married | 60 | 44 (73.3%) | 3 (5.0%) | 13 (21.7%) | |
| Married or living as married | 243 | 166 (68.3%) | 31 (12.8%) | 46 (18.9%) | |
| Formerly married/separated | 50 | 40 (80.0%) | 3 (6.0%) | 7 (14.0%) | |
| Missing | 0 | 0 | 0 | 0 | |
| **Health Insurance** | | | | | 0.49 |
| None | 68 | 52 (76.5%) | 4 (5.9%) | 12 (17.6%) | |
| Private | 139 | 99 (71.2%) | 18 (12.9%) | 22 (15.8%) | |
| Public | 128 | 85 (66.4%) | 18 (14.1%) | 22 (17.2%) | |
| Unsure | 18 | 14 (77.8%) | 0 | 4 (22.2%) | |
| **Independent variables** | | | | | |
| **COVID Test/Results** | | | | | 0.89 |
| Tested positive | 16 | 13 (81.3%) | 2 (12.5%) | 1 (6.3%) | |
| Tested negative | 81 | 58 (71.6%) | 9 (11.1%) | 14 (17.3%) | |
| Test results unknown | 1 | 1 (100%) | 0 | 0 | |
| Not been tested | 254 | 178 (70.1%) | 26 (10.2%) | 50 (19.7%) | |
| Missing | 1 | 0 | 0 | 1 (100%) | |
| **Significance of Pandemic** | | | | | **0.000** |
| Major | 275 | 209 (76.0%) | 18 (6.5%) | 48 (17.5%) | |
| Minor | 78 | 41 (52.6%) | 19 (24.4%) | 18 (23.1%) | |
| Missing | 0 | 0 | 0 | 0 | |
| **Trusted Sources of Information** | | | | | **0.001** |
| Healthcare providers | 189 | 139 (73.5%) | 14 (7.4%) | 36 (19.0%) | |
| Public agencies | 56 | 43 (76.8%) | 3 (5.4%) | 10 (17.9%) | |
| Social Networks | 22 | 14 (63.6%) | 7 (31.8%) | 1 (4.5%) | |
| Private news sources | 44 | 28 (63.4%) | 10 (22.7%) | 6 (13.6%) | |
| Missing | 42 | 26 (61.9%) | 3 (7.1%) | 13 (31.0%) | |

*Total varies due to missing responses; percentages may not total 100% due to rounding

*"Missing" refers to the number of people who were asked the question but did not respond

*For questions where participants could select multiple answers, the percentage was calculated using the number of people asked the question. For these questions, the percentage will not total 100%

*Defined as Umbanda, Candomblé, Espiritist

**Table 2. Multinomial logistic regression comparing those who did versus those who did not intend to vaccinate (N = 351).**

| | Intend to vaccinate vs do not intend to vaccinate | Intend to vaccinate vs do not intend to vaccinate |
|---|---|---|
| | **Model 1** | **Model 2** |
| | **(socio-demographic characteristics only)** | **(socio-demographic characteristics plus independent variables)** |
| | **AOR (95% CI)** | **AOR (95% CI)** |
| **Age** | 0.02 (-0.03, 0.07) | 0.03 (-0.02, 0.08) |
| **Marital Status** (ref. single) | | |
| Married or living as married | -1.23 (-2.62, 0.16) | -1.24 (-2.79, 0.29) |
| Formerly married/separated | 0.21 (-1.58, 2.00) | -0.33 (-2.27, 1.16) |
| **Income** (ref. $15,000-$25,000) | | |
| Less than $10,000 | -0.27 (-1.52, 0.99) | -0.06 (-1.42, 1.31) |
| $25,001 - $50,000 | 0.83 (-0.52, 2.17) | 1.37 (-0.13, 2.87) |
| $50,001- $75,000 | 1.24 (-0.13, 2.60) | 1.29 (-0.19, 2.77) |
| $75,001-$100,000 | 0.49 (-0.78, 1.77) | 0.77 (-0.73, 2.26) |
| Higher than $100,000 | 0.25 (-0.93, 1.44) | 0.10 (-1.20, 1.41) |
| Don't know | 1.28 (-0.94, 3.51) | 1.37 (-0.91, 3.65) |
| **Time in US** (continuous) | **-0.10 (-0.16, -0.04)** | **-0.12 (-0.19, -0.05)** |
| **Significance of Pandemic** (ref. minor) | | |
| Major | | **1.52 (0.62, 2.42)** |
| **Trusted Sources of Health Information** (ref. healthcare provider) | | |
| Public agency | | 0.10 (-1.29, 1.50) |
| Social networks | | **-1.94 (-3.25, -0.63)** |
| Private news source | | **-1.71 (-2.78, -0.63)** |
| Missing | | 0.31 (-1.14, 1.77) |

AOR = Adjusted odds ratio; 95% CI = 95% confidence intervals

health information were social networks (i.e., social media, family or friends) or private news sources (i.e., network news and internet) were significantly more likely to report that they would not be vaccinated compared with those who reported healthcare professionals as their most trusted sources of health information (social networks OR: -1.94, 95% CI:-3.25, -0.63; private news sources OR: -1.71, 95% CI: -2.78, -0.63). Trust in information from public agencies was not statistically different from trust in healthcare providers (OR: 0.10, 95% CI: -1.29, 1.50). Participants who had lived longer in the U.S. were significantly less likely to intend to get the vaccine compared to those that had lived in the U.S. for longer periods (OR: -0.12, 95% CI: -0.19, -0.05). See Table 2. We did not find differences in findings when we ran the same multivariable models comparing those who reported that they would be vaccinated versus those who reported that they were unsure (not shown).

## Discussion

Most of the Brazilian immigrant women in this study reported that they would accept a COVID-19 vaccine. In multivariable analyses controlling for a range of socio-demographic factors, those who perceived the pandemic to be a major crisis and those who most trusted information from healthcare providers were more likely to report that they would take the vaccine. Additionally, we found evidence that longer periods of time spent living in the U.S. were associated with being less likely to intend to vaccinate.

To date, few studies have assessed COVID-19 vaccine intentions among immigrant populations in the U.S. and results have been mixed, perhaps due to differences in the time when data were collected or sampling methods. Data from a nationally representative sample of N = 1936 adults ages 18–65 collected between February 12—March 3, 2021, found that COVID-19 vaccine hesitancy was similar between foreign-born Hispanics and all other (non-foreign) racial and ethnic groups [36]. However, a national survey conducted in May 2021 by the Kaiser Family Foundation found that 31% of "potentially undocumented" Hispanic adults reported having received at least one dose of the COVID-19 vaccine, compared with 61% of legal permanent residents and 46% of permanent residents [37]. Another survey conducted in December 2020 by the Urban Institute among California residents found that adults in immigrant families were more likely than other adults in to report they would definitely or probably get a COVID-19 vaccine (75% versus 68%) [38].

Our finding that over 70% of Brazilian immigrant women intended to be vaccinated is aligned with national polls among the general U.S. population taken around the same time period which reported that 50–74% of those polled intended to be vaccinated [21, 39–42]. Polls and surveys have generally found that individuals more likely to accept a vaccine were non-Hispanic White and had higher levels of education and income [40, 43]. It is possible that our finding of relatively high vaccine intentions among our sample may be related to the high level of vaccine acceptance in Brazil. For example, 73% of adults aged 60 and older living in Brazil received a flu vaccine in 2015–2016 [44], compared with only 63% of U.S adults of the same age during the same year [45]. In a large nationally representative survey done in Brazil in August 2020, 89% of respondents said that they intended to get the COVID-19 vaccine when it became available [46]. The high level of vaccine acceptance in Brazil (as opposed to the U.S.) may also help to explain our finding that the number of years living in the U.S. was negatively associated with the intention to get the vaccine.

Our observation that the perceived severity of the pandemic was associated with vaccine intentions is not surprising and is predicted by the Health Belief Model. In general, perceived severity of illness has been associated with vaccine uptake of the human papillomavirus (HPV) and influenza vaccines [47–49]. In addition, Reiter, et al. (2020) found that greater perceived severity of the pandemic was associated with an increased likelihood of accepting the COVID-19 vaccine in a large U.S. sample [21]. We also found that those who had high levels of trust in health information from healthcare providers were more likely to have positive vaccine intentions, while those who trusted social networks and private news sources were more likely to report that they would not get the vaccine or were unsure [50, 51]. Other studies have similarly found that physicians are trusted sources of health information and can serve as important cues to action to promote COVID-19 vaccine uptake among general populations [21, 34, 40, 41], and we found this in a prior qualitative study among Brazilian immigrants [52]. To strengthen the impact of health information among the Brazilian population, healthcare providers should ensure health communication regarding COVID-19 vaccination is culturally competent (e.g., tailored to meet people's social, cultural, and linguistic needs) [8, 53]. Family and friends can also be important sources of information and cues to action, although if misinformed, can perpetuate myths and misinformation [54, 55].

Before discussing the implications of our findings, it is important to acknowledge limitations. We analyzed data from a convenience sample and the survey was conducted as part of a larger study on Brazilian women's health experiences. As a result, the results may not be generalizable since those willing to complete health surveys may be more receptive to health interventions [56]. Moreover, since a monetary incentive was provided, we cannot rule out the possibility that surveys were taken by those who were ineligible but misrepresented themselves to gain the incentive. Also, since these analyses are cross-sectional, temporal relationships

between vaccine intentions and independent variables cannot be inferred. In addition, we collected this data before any COVID-19 vaccines were available, so there was no information available about vaccine characteristics (e.g., efficacy, number of doses), which may influence vaccine intentions. More research on current vaccine intentions in the Brazilian immigrant community (including men) will be necessary to update and validate our findings, especially as the pandemic continues.

Regardless, as the first study of COVID-19 vaccine intention among Brazilian immigrants in the U.S., our results can help to inform strategies to promote vaccination. While our findings suggest that many Brazilian immigrant women are amenable to taking the COVID-19 vaccine, about a third were either unsure or stated they would not be vaccinated. To reach those who were unsure or do not intend to be vaccinated, the serious nature of the pandemic should be stressed to more accurately aligned with perceived severity, although this must be done in ways that avoid causing undue stress or panic. Vaccine messaging by healthcare providers may be effective with this population. Encouragement of vaccination (i.e., 'cue to action') by healthcare providers is especially important given the strong impact of provider recommendations documented in numerous studies [57]. Specifically, studies of other vaccine types found that presumptive language by providers, which presents vaccine uptake as the default option (e.g., "We are going to administer your vaccine today"), is more effective than conversational language that actively questions the patient about their willingness to be vaccinated [57]. However, many Brazilian immigrants experience major barriers to access of health care, as they often do not have a primary source of healthcare and lack health insurance [58]. An alternative model is the dissemination of information through local public health agencies. For example, the use of community health workers that are trained and hired by public health agencies to provide vaccine information has been recommended by the World Health Organization and others [59, 60], and this model has been successfully employed in the U.S. [61]. In our prior research, we found that many Brazilians have strong family/friend ties and that these communication networks are strong [62], which suggests promise for this intervention model.

## Conclusions

Our findings indicate that most Brazilian immigrant women are amenable to taking the COVID-19 vaccine. However, to maximize the full benefits of vaccination within this community, it will be necessary to make concerted efforts to reach those who are unsure or opposed to taking the vaccine. Consistent with the Health Belief Model, our findings suggest that vaccination efforts should stress the severity of the COVID-19 pandemic and target Brazilian immigrants who have lived a long time in the U.S. Healthcare providers may be particularly suited to provide cues to action, given high levels of trust. Local cultural media outlets may also help to ensure that people have accurate information about the vaccine. While these findings provide new insights into potential COVID-19 vaccine uptake among immigrant communities in the U.S., additional research will be needed to assess prospective relationships between COVID-19 vaccination with perceived severity of the pandemic and health information from trusted information sources. Moreover, it may be useful to examine additional constructs from the Health Belief Model that were not assessed in this study, including perceived susceptibility, perceived benefits of vaccination, as well as self-efficacy in navigating the healthcare system to obtain a vaccine.

## Acknowledgments

We thank Stacy Chen and Amy Kaplan for their work on the survey. We thank Cristiane Soares for her contributions to the study. We are indebted to our community partners for collaborating with us on the study.

## Author Contributions

**Conceptualization:** Jennifer D. Allen, Leticia Priebe Rocha, Raviv Rose, Adriana Fernandes, Heloisa Galvão.

**Data curation:** Leticia Priebe Rocha, Raviv Rose, Adriana Fernandes.

**Formal analysis:** Annmarie Hoch, Thalia Porteny.

**Funding acquisition:** Jennifer D. Allen.

**Investigation:** Jennifer D. Allen, Leticia Priebe Rocha, Raviv Rose.

**Methodology:** Jennifer D. Allen, Leticia Priebe Rocha, Raviv Rose, Adriana Fernandes, Heloisa Galvão.

**Project administration:** Jennifer D. Allen, Leticia Priebe Rocha, Raviv Rose, Adriana Fernandes.

**Software:** Annmarie Hoch, Thalia Porteny.

**Supervision:** Jennifer D. Allen.

**Writing – original draft:** Jennifer D. Allen, Leticia Priebe Rocha, Raviv Rose, Annmarie Hoch.

**Writing – review & editing:** Jennifer D. Allen, Leticia Priebe Rocha, Raviv Rose, Annmarie Hoch, Thalia Porteny, Adriana Fernandes, Heloisa Galvão.

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
