## [Decision Letter · Decision Letter 0]

26 Oct 2021

PONE-D-21-22345

Intention to Obtain a COVID-19 Vaccine Among Brazilian Immigrant Women in the U.S.: Results from a Nationwide Survey

PLOS ONE

Dear Dr. Allen,

Thank you for submitting your manuscript to PLOS ONE. After careful consideration, we feel that it has merit but does not fully meet PLOS ONE’s publication criteria as it currently stands. Therefore, we invite you to submit a revised version of the manuscript that addresses the points raised during the review process.

ACADEMIC EDITOR: Considering my own reading and the lone reviewer opinion, I will recommending a major revision for this paper. 

We look forward to receiving your revised manuscript.

Kind regards,

Srinivas Goli, Ph.D.

Academic Editor

PLOS ONE

2. Please provide additional details regarding participant consent. In the Methods section, please ensure that you have specified (1) whether consent was informed and (2) what type you obtained (for instance, written or verbal). If your study included minors, state whether you obtained consent from parents or guardians. If the need for consent was waived by the ethics committee, please include this information.

3.Thank you for stating the following in the Acknowledgments/ Funding Section of your manuscript:

“Funding for this study was provided in part by a grant from the Tisch College Community Research Center. Additional funding was provided by the National Center for Advancing Translational Sciences, National Institutes of Health, Award Number UL1TR002544. The content is solely the responsibility of the authors and does not necessarily represent the official views of the NIH.”

“Funding for this study was provided to JA in part by a grant from the Tisch College Community Research Center (https://tischcollege.tufts.edu/research/tcrc). Additional funding was provided by the National Center for Advancing Translational Sciences, National Institutes of Health, Award Number UL1TR002544 (https://ncats.nih.gov/). The content is solely the responsibility of the authors and does not necessarily represent the official views of the NIH. The funders had no role in study design, data collection and analysis, decision to publish, or preparation of the manuscript.”

:

Additional Editor Comments:

Considering my own reading and the lone reviewer opinion, I will recommending a major revision for this paper.

Reviewers' comments:

Reviewer's Responses to Questions

**Comments to the Author**

1. Is the manuscript technically sound, and do the data support the conclusions?

Reviewer #1: Partly

2. Has the statistical analysis been performed appropriately and rigorously? 

Reviewer #1: No

3. Have the authors made all data underlying the findings in their manuscript fully available?

Reviewer #1: No

4. Is the manuscript presented in an intelligible fashion and written in standard English?

Reviewer #1: Yes

5. Review Comments to the Author

Reviewer #1: On data sharing question/policy: In one place the authors say the data are publicly available, in the other, they say data are not available (see last point in my reviewer comments).

Other comments for the authors:

This cross-sectional national online survey of Brazilian women describes attitudes and intentions towards acceptability of a COVID-19 vaccine. While the survey was conducted prior to the emergency approval of the current vaccines, many of the findings may still be useful as the pandemic continues to surge in the face of ongoing vaccine hesitancy. The sample is not representative, but still may provide insights into immigrant attitudes towards the vaccines. However, the paper could be strengthened by addressing the following points, should the authors choose to revise the manuscript.

• Given the timing of this survey in the context of fast-changing epidemic conditions, I suggest including the dates of the survey in the abstract so readers know right away the referent period for the study.

• In the background, some ambiguity exists between the terms “Brazilians” and “immigrant Brazilians.” For example, The authors state there are over 450,000 Brazilian in the US according to Census (L 66). If this is based on the race question, then that may not mean all of these are immigrants. Could the authors clarify? And could they be consistent in use of terms throughout?

• Relatedly, Table 1 indicates 11 respondents were born in the US. These cases should be removed from all analyses.

• Eligibility includes Brazilian women who resided in the US. How did the researchers determine immigrant status?

• Most online surveys offering compensation are the target of fraudsters attempting to access the compensation by providing false information. What safeguards were in place to ensure such attempts were blocked?

• Since vaccine hesitancy appears to follow certain geographic boundaries, it may be useful to include geography in the multivariate analysis.

• A number of questions arose based on the results/Table 2:

o The selection for the final models is purely empirically driven, not theoretically driven (L 148). Yet it appears the Health Belief Model (or perhaps a related framing) underlies the selection of questions, analysis, and discussion. Are there conceptual framings that may better inform the final models besides just empirical bivariate results?

o The selection of demographic characteristics in Model 1 is confusing. None of them were significantly related to vaccine intentions (Table 1), but somehow the authors landed on Age, marital status and income to include. Why?

o Table 2 says that n=364, same as Table 1. However, n=42 cases were missing in the “Trusted Source” variable. What happened to these cases in Model 2? If they were dropped from Model 2, then those same cases should be dropped from Model 1 – otherwise there may be systematic differences in the two samples driving results. Other strategies include imputing missing cases (in general, a better option than dropping cases), or treating them as another category of the variable (similar to what was done for income).

o The interpretation of “Trusted Source” appears to be somewhat off. Although not explicit on the table (and it should be), the referent category for the variable is “Health care provider.” The results, thus, are relative to that category, so that public agencies as the most trusted news source are not significantly different from health care providers for vaccine intention. This interpretation changes the discussion and conclusions. Note – the results themselves may change once the missing data in this variable are appropriate accommodated.

• Finally, the statements about data sharing appear to be in conflict. On the front material, the statement reads data cannot be shared publicly. In the backend topics, the authors declare that data are available on request. Which is it?

6. PLOS authors have the option to publish the peer review history of their article (what does this mean?). If published, this will include your full peer review and any attached files.

Reviewer #1: No

---

## [Author Response · Author response to Decision Letter 0]

9 Mar 2022

Please see attached cover letter and response to reviewers documents

---

## [Decision Letter · Decision Letter 1]

10 Jul 2022

PONE-D-21-22345R1Intention to obtain a COVID-19 vaccine among Brazilian immigrant women in the U.S.PLOS ONE

Dear Dr. Allen,

Thank you for submitting your manuscript to PLOS ONE. After careful consideration, we feel that it has merit but does not fully meet PLOS ONE’s publication criteria as it currently stands. Therefore, we invite you to submit a revised version of the manuscript that addresses the points raised during the review process.

The reviewer from the previous round has reassessed the manuscript. They are overall happy with the amendments made, and have provided some additional suggestions to strengthen the manuscript, which we wanted to give you the opportunity to consider.

When you resubmit, please also provide the IRB approval number for the study in the Methods section.

We look forward to receiving your revised manuscript.

Kind regards,

Hanna Landenmark

Staff Editor

PLOS ONE

Journal Requirements:

Reviewers' comments:

Reviewer's Responses to Questions

**Comments to the Author**

1. If the authors have adequately addressed your comments raised in a previous round of review and you feel that this manuscript is now acceptable for publication, you may indicate that here to bypass the “Comments to the Author” section, enter your conflict of interest statement in the “Confidential to Editor” section, and submit your "Accept" recommendation.

Reviewer #1: (No Response)

2. Is the manuscript technically sound, and do the data support the conclusions?

Reviewer #1: Partly

3. Has the statistical analysis been performed appropriately and rigorously? 

Reviewer #1: Yes

4. Have the authors made all data underlying the findings in their manuscript fully available?

Reviewer #1: Yes

5. Is the manuscript presented in an intelligible fashion and written in standard English?

Reviewer #1: Yes

6. Review Comments to the Author

Reviewer #1: The authors have been responsive to comments, and the paper is improved. I have only three comments:

1. On line 277, the word should be regarding, not regrading

2. On Table 2, the Time in US variable is a bit puzzling. The referent is 0-4 years, but no other category appears for that variable in the table, as one would expect given the categorization of the variable in Table 1. Was this variable dichotomized in the multivariate analysis? Or are the other categories missing from the table? Or was this variable used as a continuous variable, and the reference notation a mistake?

3. The authors describe the HBM as a guiding frame in the Methods, but do not return to it in the Discussion. Assessing the usefulness (or not) of the model for this investigation would be an important theoretical contribution.

7. PLOS authors have the option to publish the peer review history of their article (what does this mean?). If published, this will include your full peer review and any attached files.

Reviewer #1: No

---

## [Author Response · Author response to Decision Letter 1]

18 Aug 2022

Response to review  

We appreciate the thoughtful comments and questions addressed by the reviewers. We’re confident that the suggestions they have offered will improve the paper. Below, we have responded to all the points raised by each reviewer. 

Reviewer #1: The authors have been responsive to comments, and the paper is improved. I have only three comments:

1. On line 277, the word should be regarding, not regrading

Author response: Thank you for catching this error. We have made the change from “regrading” to “regarding.”

2. On Table 2, the Time in US variable is a bit puzzling. The referent is 0-4 years, but no other category appears for that variable in the table, as one would expect given the categorization of the variable in Table 1. Was this variable dichotomized in the multivariate analysis? Or are the other categories missing from the table? Or was this variable used as a continuous variable, and the reference notation a mistake?

Author response: Thank you for bringing this to our attention. The variable was treated as continuous, and the reference notation in the table was in error. We have corrected this in Tables 1 and 2. Changes to the manuscript text to reflect this correction have also been made (see lines 237-239). 

3. The authors describe the HBM as a guiding frame in the Methods, but do not return to it in the Discussion. Assessing the usefulness (or not) of the model for this investigation would be an important theoretical contribution.

Author response: Thank you for this suggestion. We have added language to the discussion section to highlight where our findings were consistent with the Health Belief Model. Additionally, we comment and make recommendations about the utility of the model for investigating COVID-19 vaccine behaviors (see lines 279-280,287-288, 334-345).

Editor comment

When you resubmit, please also provide the IRB approval number for the study in the Methods section.

Author response: This has been included (see line 118).

---

## [Editor Report · Decision Letter 2]

7 Sep 2022

Intention to obtain a COVID-19 vaccine among Brazilian immigrant women in the U.S.

PONE-D-21-22345R2

Dear Dr. Allen,

We’re pleased to inform you that your manuscript has been judged scientifically suitable for publication and will be formally accepted for publication once it meets all outstanding technical requirements.

Kind regards,

Carol E. Kaufman

Guest Editor and Prior reviewer for this manuscript

PLOS ONE
---

## [Editor Report · Acceptance letter]

12 Sep 2022

PONE-D-21-22345R2 

Intention to obtain a COVID-19 vaccine among Brazilian immigrant women in the U.S. 

Dear Dr. Allen:

I'm pleased to inform you that your manuscript has been deemed suitable for publication in PLOS ONE. Congratulations! Your manuscript is now with our production department. 

Kind regards, 

on behalf of

Dr. Carol E. Kaufman 

Guest Editor

PLOS ONE